# Sld3CBD–Cdc45 structural insights into Cdc45 recruitment for CMG complex formation during DNA replication

Hao Li[1], Izumi Ishizaki[1], Koji Kato[1,2†], Xiaomei Sun[2], Sachiko Muramatsu[3], Hiroshi Itou[3‡], Toyoyuki Ose[1,2], Hiroyuki Araki[3*], Min Yao[1,2*]

[1]Graduate School of Life Science, Hokkaido University, Sapporo, Japan; [2]Faculty of Advanced Life Science, Hokkaido University, Sapporo, Japan; [3]National Institute of Genetics, Mishima, Japan

*For correspondence:
hiaraki@nig.ac.jp (HA);
yao@castor.sci.hokudai.ac.jp
(MY)

Present address: †Research
Institute for Interdisciplinary
Science and Graduate School of
Natural Science and Technology,
Okayama University, Okayama,
Japan; ‡Chiome Bioscience, Inc,
Shibuya-ku, Tokyo, Japan

Reviewing Editor: Akira
Shinohara, The University of
Osaka, Japan

## eLife Assessment

This **valuable** paper describes the crystal structure of a complex of the Sld3-Cdc45-binding domain (CBD) with Cdc45, which is essential for the assembly of an active Cdc45-MCM-GINS (CMG) double-hexamer at the replication origin. The structural and biochemical analyses of protein-protein inter-actions and DNA binding provided **solid** evidence to support the authors' conclusion. The results shown in the paper are of interest to researchers in DNA replication and genome stability.

**Abstract** DNA replication requires recruitment of Cdc45 and GINS into the MCM double hexamer by initiation factors to form an active helicase, the Cdc45–MCM–GINS (CMG) complex, at the replication origins. The initiation factor Sld3 is a central regulator of Cdc45 and GINS recruitment, working with Sld7 together. However, the mechanism through which Sld3 regulates CMG complex formation remains unclear. Here, we present the structure of the Sld3 Cdc45-binding domain in complex with Cdc45 (Sld3CBD–Cdc45), showing detailed interactions and conformational changes required for binding to each other. The mutant analysis indicated that the binding between Sld3CBD and Cdc45 could be broken easily. We also revealed that Sld3CBD, GINS, and MCM bind to different sites on Cdc45 in the Sld3CBD–CMG model, indicating that after recruitment of Cdc45, Sld7–Sld3 could remain in Cdc45–MCM until CMG formation. The consistency between the particle size of Sld7–Sld3–Cdc45 and the distance between Sld3CBDs in the Cdc45–MCM dimer indicated the binding manner of the Cdc45–Sld3–[Sld7]₂–Sld3–Cdc45 off/on MCM double hexamer. A DNA-binding assay of Sld3 and its complexes with single-stranded ARS1 (autonomously replicating sequence 1) fragments revealed a relationship between the dissociation of Sld7–Sld3 from CMG and the unwound single-stranded DNA. These findings help to further our understanding of the molec-ular basis of the regulation of CMG complex formation by Sld3.

## Introduction

Eukaryotic chromosomal DNA replication in the cell cycle, a tightly regulated process that ensures precise gene copying, begins with the unwinding of double-stranded DNA (dsDNA) into single-stranded DNA (ssDNA), including the formation of an active helicase CMG complex (*Tanaka and Araki, 2013*; *Costa and Diffley, 2022*). In the G1 phase of the yeast cell cycle, two inactive helicase Mcm2–7 hexamer rings (Mcm2–6–4–7–3–5) are loaded onto autonomously replicating sequences (ARSs) of dsDNA to form a double hexamer (MCM DH) at the replication origin (*Evrin et al., 2009*; *Remus et al., 2009*; *Ticau et al., 2017*). To facilitate this loading, Cdt1 binds to the Mcm2–7 hexamer

ring and stabilizes a gap formation between the Mcm2 and Mcm5, which serves as a dsDNA entry gate (*Samel et al., 2014*; *Zhai et al., 2017*; *Bochman et al., 2008*; *Frigola et al., 2017*). Next, the MCM DH on dsDNA is phosphorylated by Dbf4-dependent protein kinase (DDK, also known as Cdc7 kinase) (*Figure 1—figure supplement 1A*), allowing the key factor Sld3 and its partner Sld7 to recruit Cdc45 (Sld7–Sld3–Cdc45) to each MCM (*Figure 1—figure supplement 1B*; *Tanaka et al., 2011*; *Heller et al., 2011*; *Kamimura et al., 2001*; *Nakajima and Masukata, 2002*). Subsequently, the phosphorylation of Sld3 by cyclin-dependent kinase (CDK) regulates GINS assembly onto MCM DH with Dpb11, CDK-phosphorylated Sld2, and DNA polymerase ε to form CMG (*Figure 1—figure supplement 1* C) (*Dhingra et al., 2015*; *Choi et al., 2007*; *Bruck et al., 2011*; *Masumoto et al., 2002*; *Tanaka et al., 2013*; *Tanaka et al., 2007*). Finally, the factors Sld2, Sld3, Sld7, and Dpb11 are released by elusive mechanisms from an active CMG, and bidirectional replication by translocating along the 3' to 5' direction of the DNA strand (*Figure 1—figure supplement 1D*) is facilitated (*Costa et al., 2011*; *Szambowska et al., 2014*).

As a central regulator of CMG formation in *Saccharomyces cerevisiae*, Sld3 functions as a bridge protein in the Sld7–Sld3–Cdc45 complex to recruit Cdc45 to DDK-phosphorylated MCM DH. This Sld3 contains three domains, each of which binds mainly to one of Sld7, Cdc45, and MCM. The N-terminal domain of Sld3 (Sld3NTD: M1–L116) binds to the N-terminal domain (M1–D130) of Sld7 (Sld7NTD) (Sld7 C-terminal domain [Sld7CTD: K168–S257] is a self-dimerization domain) (*Itou et al., 2015*). Following the NTD of Sld3, the middle part is a central portion, the Cdc45-binding domain (Sld3CBD: S148–K430) (*Kamimura et al., 2001*; *Itou et al., 2014*). A previous study also demonstrated that a region (L510–R530) in the Sld3 C-terminal domain (Sld3CTD: T445–T679) binds to Mcm4 and 6 (*Deegan et al., 2016*). CDK-dependent phosphorylation of two residues (T600 and S622) downstream of Sld3CTD initiates the binding of GINS-carrying Dpb11 to Cdc45–MCM (*Tanaka et al., 2007*; *Zegerman and Diffley, 2007*). In addition to its role as a bridge in the recruitment of Cdc45 and GINS, Sld3 binds to two single-stranded DNA fragments of ARS1 (ssARS) identified as the origin of DNA replication (ssARS1-2 and ssARS1-5), but not to the corresponding double-stranded ARS1 (dsARS) (*Bruck and Kaplan, 2011*). This specific Sld3–ssDNA association is not affected by CDK phosphorylation. Furthermore, the structures of the Sld7CTD dimer (PDBID:3X38) (*Itou et al., 2015*), Sld7NTD–Sld3NTD (PDBID:3X37) (*Itou et al., 2015*), Sld3CBD (PDBID:3WI3) (*Itou et al., 2014*), MCM DH (6F0L) (*Abid Ali et al., 2017*), CMG (PDBIDs:3JC5, 3JC6, 3JC7) (*Yuan et al., 2016*), CMG-DNA-polε (PDBID:7Z13) (*Lewis et al., 2022*), CMG-DONSON-DNA (PDBID:8Q6O) (*Cvetkovic et al., 2023*), and so forth, have been determined through crystallography and cryogenic electron microscopy (cryo-EM). Cdc45 belongs to the DHH superfamily of proteins defined by the conserved triad motif DHH (Asp-His-His), and contains a DHH-associated domain (DHHA1: R523–L650) at its C-terminus (*Simon et al., 2016*). Recent single-molecule biochemical assays have reported the stepwise recruitment of multiple Cdc45s to the MCM DH (*De Jesús-Kim et al., 2021*). However, how Sld3–Sld7 recruits Cdc45 onto the MCM for CMG formation to regulate the initiation of DNA replication remains unclear.

The present study aimed to understand how Sld3 recruits Cdc45 to the MCM DH with Sld7 for CMG formation through structure and particle analyses. We determined the structure of *S. cerevisiae* Sld3CBD–Cdc45 at 2.6 Å resolution and presented the detailed interactions between Sld3 and Cdc45, confirmed through in vitro and in vivo mutant analyses. Compared to the monomer structures, the conformation of Sld3CBD and Cdc45 in the Sld3CBD–Cdc45 complex changed significantly for binding to each other. Based on the structural similarity of Cdc45 in Sld3CBD–Cdc45 and CMG, we modelled Sld3CBD–Cdc45–MCM–dsDNA and SCMG–dsDNA (Sld3CBD–CMG–dsDNA) as a snapshot of how helicase CMG forms. The models demonstrated that Sld3CBD, MCM, and GINS bind to different sites on Cdc45, indicating that Sld7–Sld3 could remain at the Cdc45–MCM until CMG formation after GINS loading. Consistency between the particle size of Sld7–Sld3ΔC–Cdc45 (Sld3ΔC: M1–K430; truncated the C-terminal domain) as per spectroscopic analysis, and the distance of Sld3CBDs in the Cdc45–MCM dimer suggested that the ternary complex of Sld7–Sld3–Cdc45 forms a dimer off/on the MCM DH for recruiting Cdc45. Furthermore, ssDNA-binding analysis of ARS1 fragments suggested that the release of Sld3–Sld7 could be associated with ssARS1, unwound by CMG. Our findings illustrate the recruit–release function of Sld3 in CMG formation, expanding our knowledge of the initiation process of DNA replication.

# Results

## The overall structure of Sld3CBD and Cdc45 complex

We obtained a recombinant Sld3CBD–Cdc45 complex of *S. cerevisiae* and determined its crystal structure at 2.6 Å resolution using molecular replacement (*Figure 1A*, *Figure 1—figure supplements 2A and 3*). Only one complex molecule Sld3CBD–Cdc45 exists within an asymmetric unit. Similar to the monomeric Sld3CBD (PDBID: 3WI3, with 0.50 Å RMSD for 181 Cα atoms) (*Itou et al., 2014*), Sld3CBD (Y154–P420) in the Sld3CBD–Cdc45 complex is an α-helical structure with two disordered regions (R317–S336 and P364–A369). Interestingly, a disordered part in monomeric Sld3CBD was visualized as a C-terminal part of long bent helix α8 (F294–R316; hereafter referred to as α8CTP) (*Figure 1—figure supplements 3 and 4A*, *Figure 1—figure supplement 5*). Cdc45 (M1–L650) is an α/β structure composed of three β-sheets (anti-parallel: β1-β6-β5-β4-β2-β3, anti-parallel: β7-β8, mixed: β9-β10-β11-β13-β12) surrounded by 21 α-helices (*Figure 1—figure supplement 3*). Owing to the poor electron density map, the regions D106–K110, D166–K227, S306–V310, and T438–D460 on the molecular surface could not be built. In contrast to the monomeric human Cdc45 (huCdc45) and the CMG form (Cdc45 in the yeast CMG complex), the N-terminal part of the protruding long helix α7 D219–H231 was disordered in the Sld3CBD–Cdc45 complex (*Figure 1—figure supplements 4B and 6*).

## Conformational changes in Sld3CBD and Cdc45 for binding to each other

Sld3CBD binds to Cdc45 in a way similar to a toothed gear (*Figure 1B*), with a contact surface of 1808 Å² accounting for approximately 13.3% and 7.0% of the total surface of Sld3CBD and Cdc45, respectively. Two helices (α8 and α9) of Sld3CBD formed a plier structure and gripped the C-terminal domain DHHA1 (R523–L650) of Cdc45 through hydrophobic interactions and hydrogen bonds. The C-terminal loop (L646–L650) and the C-terminal part of α7 (E232–S242) of Cdc45 sandwiched the helix α8CTP of Sld3CBD. Compared to the isolated forms (PDBIDs: 5DGO and 6CC2 for huCdc45; *Simon et al., 2016*) and EhCdc45 (*Kurniawan et al., 2018*, respectively) and the CMG form (PDBID: 3JC6; *Yuan et al., 2016*), the Cdc45 in the Sld3CBD–Cdc45 complex changed the conformation of DHHA1 to form pockets on its two sides for binding Sld3CBD α8CTP and α9. The remaining part of Cdc45 (~K517) retained a structure with RMSD values of 1.29 Å (isolated huCdc45), 1.81 Å (isolated EhCdc45), and 1.295 Å (in CMG) for 243, 251, and 361 Cα atoms, respectively.

The Sld3CBD helix α8CTP (F294–R316), surrounded by the C-terminal domain DHHA1 and a C-terminal part of the protruded long helix α7 (E232–S242) of Cdc45 (*Figure 1B*), seems to be an intrinsically disordered segment. When Sld3 is alone, it is disordered but folds into a visualizable helix coupled to the binding partner Cdc45 in the Sld3CBD–Cdc45 complex (*Figure 1—figure supplement 4A*). Previous studies reported that this α8CTP is essential for binding to Cdc45, as its deletion inhibited cell growth (*Itou et al., 2014*). Furthermore, proline substitution for Cdc45 Ser242 (strain Cdc45-35; *Kamimura et al., 2001*), which interacts with L307 and T310 in Sld3CBD α8CTP (*Figure 1C*), conferred temperature-sensitive growth to yeast cells (*Figure 2—figure supplement 1*). Although an increase in the number of copies of Sld3–Sld7 could weakly suppress cell growth defects, it did not recover the disrupted interaction.

The α9 (L337–E360) of Sld3CBD is the secondary Cdc45-binding region, which is located at a shallow dent formed by the C-terminal helix and a loop with the following β-strand (K520–Q531) of Cdc45 DHHA1 (*Figure 1B, D and E*). Three hydrophobic residues (I352, I355, and L356) in Sld3CBD α9 interact with the C-terminal sheet (L522, L527, and V529) and helix (L641 and L647) of Cdc45 DHHA1 (*Figure 1D*), while the side chains of two negatively charged residues (D344 and D348) form hydrogen bonds through a water molecule to the main and side chains of R523 on a loop of Cdc45 DHHA1 (*Figure 1E*). A disordered region, L527–V529, of Cdc45 DHHA1 in the isolated form forms a β-sheet in the Sld3CBD–Cdc45 complex and binds to I352 and I355 of Sld3CBD α9. We substituted single, double, or triple positively charged or hydrophilic residues at D344, D348, I352, I355, and L356 of Sld3CBD α9 (D344R/D348R: Sld3-2R, I352Y: Sld3-Y, I352S/I355S/L356S: Sld3-3S, I352E/I355E/L356E: Sld3-3E). Based on the CD spectra, we confirmed that these mutants retained the structural elements of Sld3CBD (*Figure 2—figure supplement 2*). Double and triple mutants Sld3-2R, Sld3-3S, and Sld3-3E eliminated Cdc45-binding affinity, whereas the single mutant Sld3-Y seemed to retain a faint interaction with Cdc45 (*Figure 2A*). We also substituted the single, double, or triple

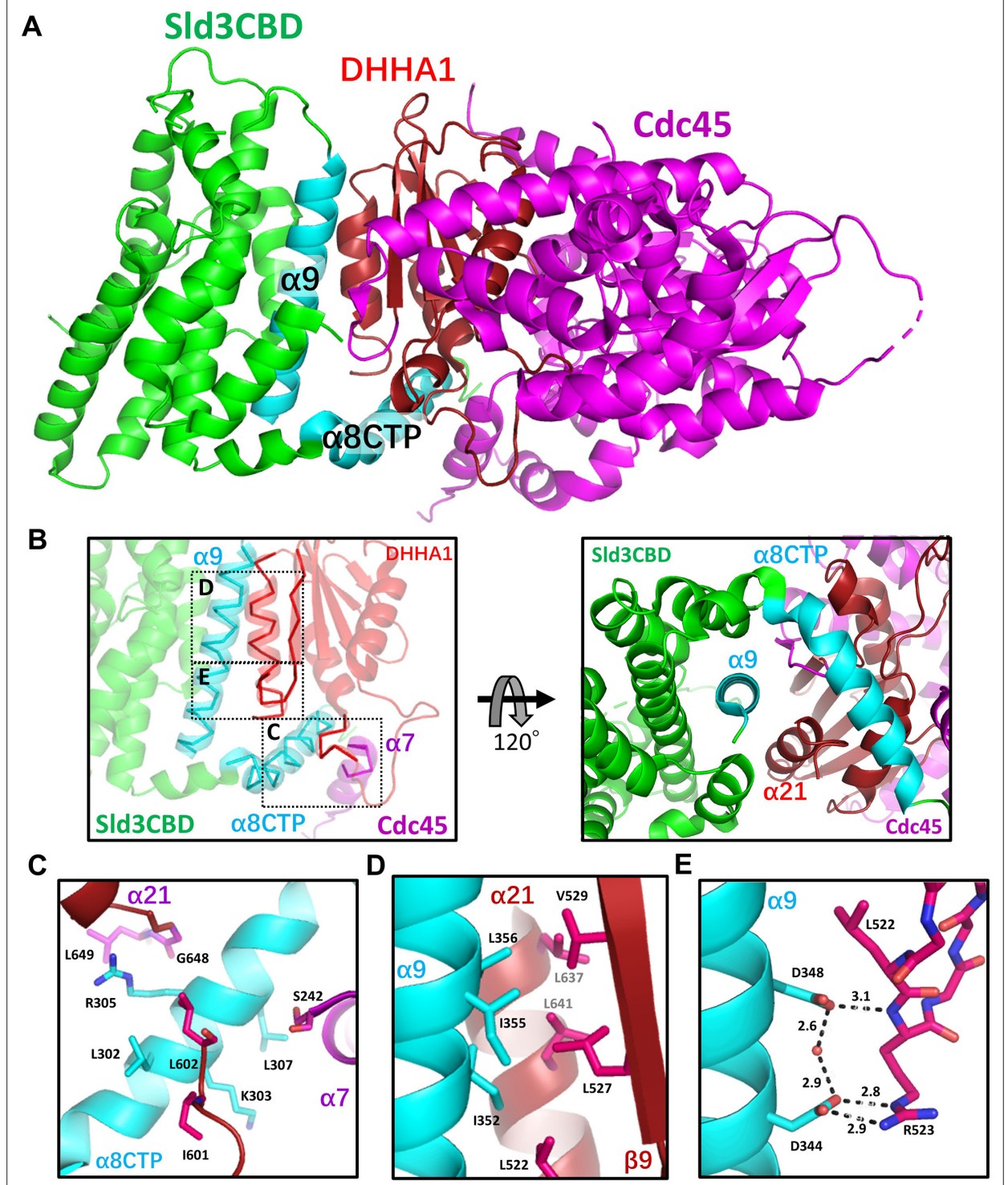

**Figure 1.** Sld3CBD–Cdc45 complex structure. (**A**) Structure of the Sld3CBD–Cdc45 complex. Sld3CBD and Cdc45 are colored in green and magenta, respectively. Cdc45-binding parts α8CTP and α9 are labelled and colored in cyan. The DHHA1 of Cdc45 is labelled and colored in dark magenta. (**B**) Binding part of Sld3CBD-Cdc45 in different viewing. Dotted squares **C**, **D**, and **E** mark three binding sites, corresponding to the bottom panel. (**C**) Binding site on the α8CTP of Sld3CBD. (**C**) (**D**) Two binding sites involving hydrophobic and hydrogen-bond interactions on the α9 of Sld3CBD, respectively. The interacting residues are depicted by sticks and labelled. The black dotted lines show hydrogen bonds.

The online version of this article includes the following source data and figure supplement(s) for figure 1:

**Figure supplement 1.** Formation of the Cdc45–MCM–GINS (CMG) complex after MCM double hexamer (DH) bound to the replication initiation site.

*Figure 1 continued on next page*

*Figure 1 continued*

**Figure supplement 2.** Purification of the Sld3CBD–Cdc45 and Sld7–Sld3ΔC–Cdc45 complexes.

**Figure supplement 2—source data 1.** PDF file containing original SDS-PAGE for *Figure 1—figure supplement 2*, indicating the relevant bands and treatments.

**Figure supplement 2—source data 2.** Original files for SDS-PAGE displayed in *Figure 1—figure supplement 2*.

**Figure supplement 3.** The topology diagram of Sld3CBD-Cdc45.

**Figure supplement 4.** Structural Comparison of Sld3CBD and Cdc45.

**Figure supplement 5.** Sequence alignment of Sld3/Treslin domain with structural elements.

**Figure supplement 6.** Sequence alignment of Cdc45s with structural elements.

residues R523, L522, L527, V529, L637, and L641 of Cdc45 on Sld3CBD α9 binding sites (L637S/L641S: Cdc45-IIS, L637E/L641E: Cdc45-IIE, L522S/L527S/V529S: Cdc45-IIIS, L522E/L527E/V529E: Cdc45-IIIE, and Cdc45-A: R523A). All Cdc45 mutants disrupted the binding between Sld3CBD and Cdc45, except for Cdc45-A, as surmised from the Sld3CBD–Cdc45 structure (*Figure 2B*). Although mutant Cdc45-A eliminated three hydrogen bonds with D344 of Sld3CBD, the remaining hydrogen-bond network maintains contact between Sld3CBD and Cdc45. Furthermore, in vivo genetic studies confirmed the importance of these Sld3 residues. Expression of Sld3-3S, Sld3-3E, and Sld3-2R in Sld3 caused no growth, while the Sld3-Y strain maintained cell growth (*Figure 2C*). These results demonstrate that the cooperative action of these residues is essential for Cdc45 binding, and loss of Sld3's Cdc45-binding affinity inhibits Cdc45 recruitment and subsequent formation of active replicative helicase CMG for DNA replication.

In comparison with Cdc45 alone (huCdc45) or CMG form (in CMG complex), domain DHHA1 of Cdc45 changed conformation significantly for binding to Sld3CBD (*Figure 1—figure supplement 4C*). The loop I595–N604 in Cdc45 DHHA1 changed conformation to interact with α8CTP of Sld3CBD. Subsequently, the helix α19 (F605–E615) rotated the C-terminus by 25 degrees, which altered the conformation of the next two β-strands (β12 and β13) in the mixed β-sheet (β9-β10-β11-β13-β12) (*Figure 1—figure supplements 3 and 4C*), allowing Sld3CBD α8CTP to enter the binding pocket. Interestingly, the Sld3CBD-Cdc45 structure shows that the Sld3CBD binding site of Cdc45 is distinct from the binding site of Cdc45 with GINS or MCM, suggesting that the Sld3CBD, Cdc45, and GINS could bind to MCM without steric clash (*Figure 2—figure supplement 3A*). Furthermore, we conducted a mutation analysis on two Cdc45 residues involved in binding to MCM (Cdc45 G367D) and GINS (Cdc45 W481R) (*Denkiewicz-Kruk et al., 2025*), respectively, and found that these mutations did not disrupt the Sld3CBD-Cdc45 complex (*Figure 2—figure supplement 3B*).

## Cdc45 recruitment to MCM DH by Sld3 with partner Sld7

Except for the Sld3 binding region DHHA1, the N-terminal domain of Cdc45 (Cdc45NTD) retained a structure similar to that in the monomer, Sld3CBD–Cdc45 complex, and CMG complex. Therefore, we modelled Sld3CBD–Cdc45–MCM–dsDNA and SCMG–dsDNA (*Figure 3*) by superposing Cdc45NTD structures between Sld3CBD–Cdc45 and each monomer of CMG dimer, which was modelled by superposing Mcm2–7 structures between CMG (PDBID: 3JC6) (*Yuan et al., 2016*) and MCM DH–dsDNA (PDBID: 5BK4) (*Noguchi et al., 2017*). In the models, two Sld3CBDs are located in each monomer of the Cdc45–MCM dimer over 230 Å apart (*Figure 3A*).

To investigate how Sld7–Sld3 brings Cdc45 to MCM DH, we attempted particle analysis for the Sld7–Sld3–Cdc45 complex using the purified recombinant Sld7–Sld3ΔC–Cdc45. The approximate molecular weight of Sld7–Sld3ΔC–Cdc45 was estimated to be >400 kDa according to a weight calibration of size-exclusion chromatography (*Figure 1—figure supplement 2B*). Given that the molecular weight calculated from its amino acid sequences was 158 kDa, the purified complex should be a dimer. Considering the Sld7–Sld3ΔC–Cdc45 dimer should be a long, rod-shaped molecule, the estimated value from SEC could be larger than the theoretical values. Subsequently, using dynamic light scattering (DLS), the particle size (hydrodynamic diameter) of the tripartite complex was estimated to be around 232 Å (*Figure 3B*, *Figure 3—figure supplement 1*), which is consistent with the distance of Sld3CBDs in the model of Sld3CBD–Cdc45–MCM dimer. To further validate the SEC and DLS results, we performed size-exclusion chromatography coupled with small-angle X-ray scattering (SEC-SAXS), which suggested a molecular weight of 370–420 kDa, and an Rg >85 Å (*Figure 3—figure supplement*

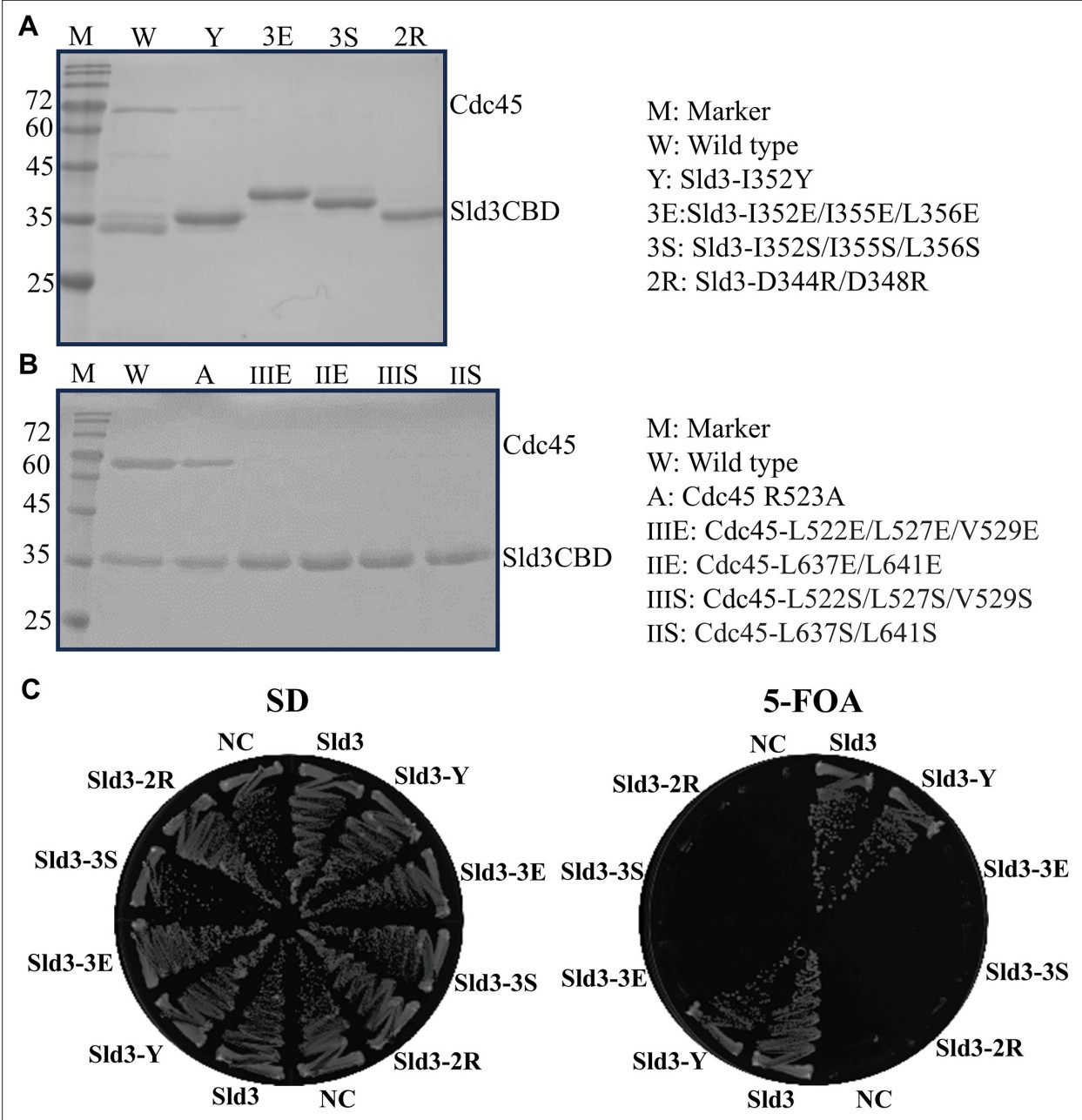

**Figure 2.** Mutation analysis of interacting residues. (**A**) In vitro binding analysis was checked using SDS-PAGE after Ni-affinity chromatography extraction of co-overexpressed Cdc45 with each Sld3 mutant. Sld3 was tagged by His-tag to bind to the column. The labels M, W, Y, 3E, 3S, and 2R are explained on the right. (**B**) In vitro binding analysis was checked using SDS-PAGE after Ni-affinity chromatography extraction of co-overexpressed Sld3 with each Cdc45 mutant. Sld3 was tagged by His-tag to bind to the column. The labels M, W, A, IIIE, IIE, IIIS, and IIS are explained on the right. (**C**) In vivo cell growth analysis of yeast cells carrying *sld3* mutations. The yeast YYK13 cells carrying *SLD3* or its mutant plasmids were streaked onto SD and FOA plates and then incubated at 298 K for 3 days. YYK13 yeast is a mutant lacking the *SLD3* gene with added *SLD3/sld3* mutant gene (YCplac22 plasmid containing *SLD3* or *sld3* mutant) that grew on SD and FOA plates. The empty plasmids were used as a negative control (NC). Mutations in Sld3-Y, Sld3-3E, Sld3-3S, and Sld3-2R are the same as those in **A**.

The online version of this article includes the following source data and figure supplement(s) for figure 2:

**Source data 1.** PDF file containing original SDS-PAGE for Figure2, indicating the relevant bands and treatments.

**Source data 2.** Original files for SDS-PAGE displayed in Figure2.

**Figure supplement 1.** In vivo mutation analysis of Cdc45 using mutant cells.

**Figure supplement 2.** SDS-PAGE analysis and circular dichroism spectra of Sld3 mutants.

*Figure 2 continued on next page*

*Figure 2 continued*

**Figure supplement 3.** Mutation analysis of Cdc45.

**Figure supplement 3—source data 1.** PDF file containing original SDS-PAGE for *Figure 2—figure supplement 3*, indicating the relevant bands and treatments.

**Figure supplement 3—source data 2.** Original files for SDS-PAGE displayed in *Figure 2—figure supplement 3*.

*2*). Considering that the domains of Sld7 (NTD: Sld3NTD-binding, CTD: self-dimerization) and Sld3 (NTD: Sld7NTD-binding, CBD: Cdc45-binding, CTD: MCM-binding) function independently (*Itou et al., 2015*; *Itou et al., 2014*), we estimated a dimer model of Sld7–Sld3ΔC–Cdc45, as shown in *Figure 3B*. Because the domains of Sld7–Sld3–Cdc45 are linked by long loops, the dimer forms a long shape with high flexibility.

Our SCMG–dsDNA model demonstrated that Sld3CBD neighbors Mcm2 and binds to Cdc45 on the opposite side of GINS binding (*Figure 3C*), indicating that Sld7–Sld3 bound to the Cdc45–MCM dimer does not contact GINS and could remain until CMG formation. A recent study also reported a structure of the DONSON (Sld2 homolog) dimer with CMG (8Q6O), showing that the DONSON dimer delivers GINS to MCM and reconfigures the MCM motors in the double CMG (*Cvetkovic et al., 2023*). The DONSON dimer is loaded at the GINS site on MCM, which is the opposite of Sld3CBD in the SCMG-dsDNA model. Interestingly, in the Sld3CBD–Cdc45–MCM–dsDNA and SCMG–dsDNA models, we found that a part (S358-D383) of Sld3CBD, containing the C-terminal of α9, a disordered fragment, and α10, was in contact distance with Mcm2CTD. In addition, the Sld3CBD-bound Cdc45 DHHA1 appeared to be close to Mcm2CTD. In contrast, the Cdc45 DHHA1 does not contact Mcm2–7 or GINS in the CMG structure (*Figure 3—figure supplement 3*). The conformational change in Cdc45 DHHA1 not only facilitates binding with Sld3CBD but could also lead to contact with Mcm2.

## ssDNA binding affinity of Sld3 depended on complex formation with Cdc45 and Sld7

Previous studies showed that Sld3 binds directly to single-strand DNA fragments (ssARS1-2 and ssARS1-5) of ARS1 identified as an origin of DNA replication (*Bruck and Kaplan, 2011*). ARS1 was divided into three 80 bp segments: ARS1-12, ARS1-34, and ARS1-56. These dsDNA segments could unwind into six single-stranded DNA fragments of 80 nucleotides (nt) in length: ssARS1-1, 2, 3, 4, 5, and 6 (*Figure 4—figure supplement 1A*). Given that Sld3 binds to Sld7 and Cdc45 on the MCM–DNA complex during CMG formation, we investigated whether Sld7 and Cdc45 affect the ssDNA-binding affinity of Sld3. Therefore, we performed an ssDNA binding assay using the Sld3CBD, Sld3CBD–Cdc45, Sld7–Sld3ΔC–Cdc45 and Sld7–Sld3ΔC (Sld3ΔC: M1–K430, domains for binding Sld7 and Cdc45) complexes against different regions of ssARS1 for comparison. Due to limitations in protein overexpression, we utilized Sld7–Sld3ΔC–Cdc45 and Sld7–Sld3ΔC from *K. marxianus* (same family as *S. cerevisiae*). Moreover, Cdc45 exhibits higher affinity for >60 nt ssDNA compared with shorter ssDNA (*Bruck and Kaplan, 2013*). Therefore, we fragmented each ssARS1 into fragments of 40 nt to prevent such nonspecific binding (*Figure 4A*, *Figure 4—figure supplement 1B*).

To investigate the specificity of the ssDNA binding affinity of Sld3, we employed an electrophoretic mobility shift assay (EMSA) using non-denaturing PAGE (native-PAGE) with ssDNAs or proteins alone as controls (*Figure 4B*, *Figure 4—figure supplement 2*). Additionally, an ssDNA (ssARS1-3_1) with no binding affinity to protein samples was used as a negative control (NC), where no ssDNA band disappeared and no new ssDNA band appeared. For the Sld3CBD and ssDNA mixtures at a molar ratio of 1:1, the band corresponding to ssDNA disappeared, indicating a binding affinity between Sld3CBD and ssDNA (*Figure 4C*). The ssDNA band remained when the Sld3CBD–Cdc45 complex was mixed with ssDNA at the same ratio, indicating that the binding affinity of Sld3CBD–Cdc45 for ssDNA was lower than that of Sld3CBD alone (*Figure 4C*). Additionally, the decrease of ssDNA bands correlated with an increase in the concentration of Sld3CBD–Cdc45 in the mixture.

In the presence of the Sld7–Sld3ΔC–Cdc45 complex, the band corresponding to ssDNA disappeared under the equimolar ratio of protein and ssDNA, similar to that observed for Sld3CBD. In this case, smeared bands appeared in the high molecular weight part of ssDNA-staining PAGE (*Figure 4B*, *Figure 4—figure supplement 2*). The positions of smeared ssDNA bands correspond to those of protein in the protein-stain pages, indicating that ssARS1 were complexed with proteins.

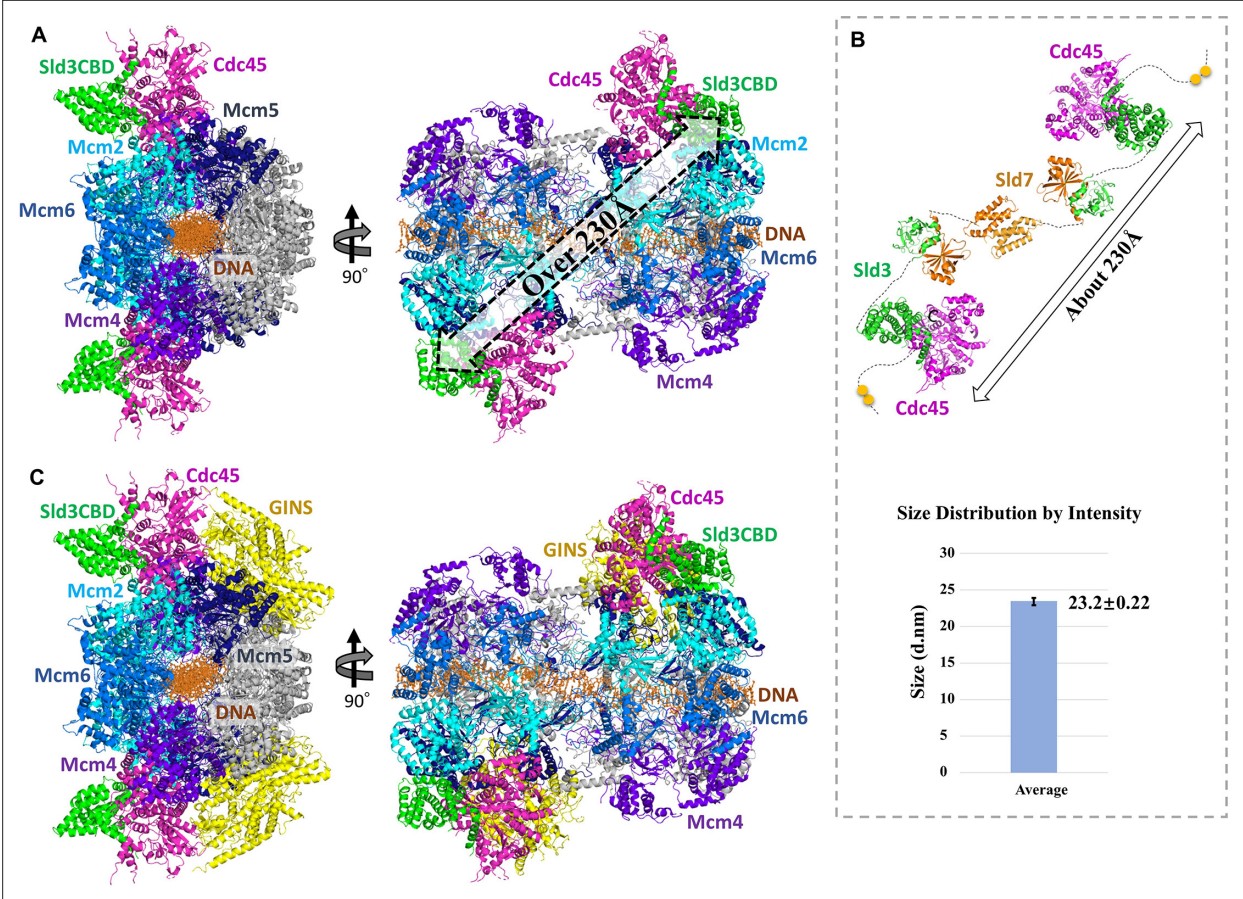

**Figure 3.** Ribbon models of complexes in dimer form and particle analysis. (**A**) Sld3CBD–Cdc45–MCM–dsDNA complex. Mcm2, 5, 4, and 6 subunits are colored in cyan, blue, marine, and light blue, respectively. Subunits Mcm3 and Mcm7 are colored in gray. Green and pink are used to color Sld3CBD and Cdc45, respectively. dsDNA is represented by a dark-orange stick. (**B**) Sld3CBD–Sld7–Cdc45 dimer before associating with the MCM DH. Sld3 and Cdc45 are shown in the same color as they are in A, while Sld7 is colored in orange. The two phosphorylated Sld3 residues are depicted as yellow balls. Particle analysis of Sld7–Sld3ΔC–Cdc45 through dynamic light scattering is shown on the bottom panel. The average peak size of the particle size distribution of the Sld7–Sld3ΔC–Cdc45 complex was estimated to be 232 Å in diameter. The measurement was carried out independently four times (*Figure 3—figure supplement 1*). (**C**) SCMG–dsDNA complex. GINS is shown in yellow, and the remainder are colored identically to those in A and B.

The online version of this article includes the following figure supplement(s) for figure 3:

**Figure supplement 1.** Dynamic light scattering (DLS) of Sld7–Sld3ΔC–Cdc45.

**Figure supplement 2.** Size-exclusion chromatography (SEC)-small-angle X-ray scattering (SAXS) analysis of Sld7–Sld3ΔC–Cdc45.

**Figure supplement 3.** DHHA1 domains of Cdc45s.

This result demonstrates that the presence of Sld7 and Sld3NTD could increase the ssDNA-binding affinity to a level comparable to that of Sld3CBD. Also, Sld7–Sld3ΔC showed the ssDNA-binding affinity similar to that of Sld7–Sld3ΔC–Cdc45, implying that the ssDNA-binding of Sld7–Sld3ΔC is independent of Cdc45. Furthermore, the results revealed no significant difference in binding affinity between the 40-base fragments of ssARS1-2-1, 2, 3 and ssARS1-5-1, 2, 3 for Sld3CBD, Sld3CBD–Cdc45, Sld7–Sld3ΔC–Cdc45, and Sld7–Sld3ΔC, indicating that there is no stronger binding-region specific to ssARS1-2 or ssARS1-5 fragments. For sequence specificity, we also analyzed other fragments (ssARS1-1, 3, 4, and 6, and dsARS1), and all of them showed no binding (*Figure 4—figure supplement 3*).

The surface charge of the Sld3CBD–Cdc45 structure shows that Cdc45 covers the main positive charge region of Sld3CBD α8CTP (*Figure 4—figure supplement 4A*), which may weaken the binding affinity of Sld3CBD–Cdc45 to ssDNA. Conversely, on the Sld3–Sld7 structure, there is a large positive charge area strip on Sld7NTD (*Figure 4—figure supplement 4B*). Considering that ssARS1 is unwound from dsARS1 by the activated helicase CMG complex formed after loading Cdc45 and

GINS, the binding affinity of Sld3–Sld7 may provide an advantage for the dissociation of Sld7–Sld3 from the CMG complex.

## Discussion

As a central regulator of helicase CMG formation, Sld3 has attracted interest in studies aiming to understand the initiation of DNA replication. Owing to the lack of structural information on Sld3 complexed with Cdc45 or MCM, how Sld3 regulates the formation of activated helicase CMG with other factors remains unknown. Here, we present the structure of the Sld3CBD–Cdc45 complex and particle size of the Sld7–Sld3–Cdc45 complex to examine the molecular mechanisms underlying CMG formation.

Sld3 exhibits high conservation across eukaryotes, whereas its functional ortholog in metazoans, Treslin (also known as TICRR), has a distinct size and sequence, except for the Cdc45-binding domain

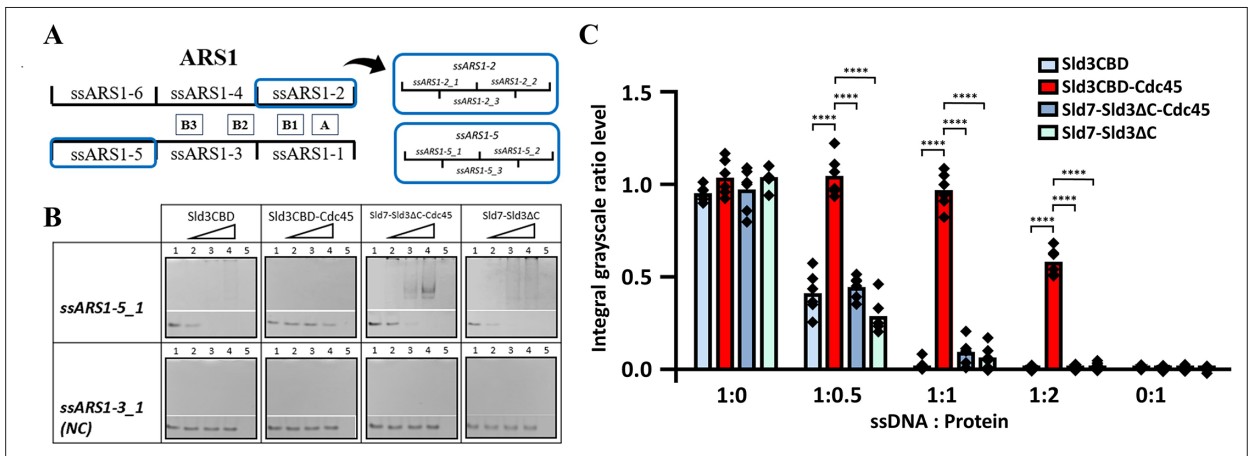

**Figure 4.** Electrophoresis mobility shift assay (EMSA) of single-stranded DNA (ssDNA) binding to Sld3 and its complexes with Sld7 and Cdc45. (**A**) Schematic of ssARS1-1–ssARS1-6. ARS1 is identified as an origin of DNA replication and divided into three 80 bp segments: ARS1-12, ARS1-34, and ARS1-56. These double-stranded DNA (dsDNA) segments could unwind into six single-stranded DNA fragments with 80 nt length: ssARS1-1, 2, 3, 4, 5, and 6. The important elements of A (autonomously replicating sequence, ARS consensus sequence), B1, B2, and B3 for unwinding are marked (**Newlon and Theis, 1993**). We divided ssARS1-2 and ssARS1-5 fragments (blue squares) into 40-base lengths for electrophoretic mobility shift assay (EMSA). (**B**) ssDNAs were visualized using Fast Blast DNA stain on polyacrylamide gels. In the presence of ssDNA fragments, Sld3CBD, Sld3CBD–Cdc45, Sld7–Sld3ΔC–Cdc45 and Sld7–Sld3ΔC were incubated with molecular mass-related concentrations. The molecular ratio of ssDNA to protein in lanes 1, 2, 3, 4, and 5 was 1:0, 1:0.5, 1:1, 1:2, and 0:1, respectively. The controls for ssDNA and protein are lanes 1 and 5, respectively. No binding ssDNA group (ssARS1-3_1) is shown at the bottom as a negative control (NC). The overall views of the EMSA results are shown in **Figure 4—figure supplement 2**. (**C**) The integral grayscale of the ssDNA bands was calculated and compared to the average of the ssDNA control band to determine the residual levels, showing differences in binding affinity. By three ratios, Sld3CBD-Cdc45 demonstrated a significantly ssDNA residual level (t-test, ****p<0.0001) compared to other samples, indicating low binding affinity to ssDNA.

The online version of this article includes the following source data and figure supplement(s) for figure 4:

**Source data 1.** PDF file containing original native-PAGE for **Figure 4**, indicating the relevant bands and treatments.

**Source data 2.** Original files for native-PAGE displayed in **Figure 4**.

**Figure supplement 1.** Sequence of ssARS1 fragments.

**Figure supplement 2.** Electrophoresis mobility shift assay (EMSA) of ssDNA binding to Sld3 and its complexes with Sld7 and Cdc45.

**Figure supplement 2—source data 1.** PDF file containing original native-PAGE for **Figure 4—figure supplement 2**, indicating the relevant bands and treatments.

**Figure supplement 2—source data 2.** Original files for native-PAGE displayed in **Figure 4—figure supplement 2**.

**Figure supplement 3.** DNA-binding assay by electrophoresis mobility shift assay.

**Figure supplement 3—source data 1.** PDF file containing original native-PAGE for **Figure 4—figure supplement 3**, indicating the relevant bands and treatments.

**Figure supplement 3—source data 2.** Original files for native-PAGE displayed in **Figure 4—figure supplement 3**.

**Figure supplement 4.** Surface charge of Sld3CBD-Cdc45 and Sld3NTD-Sld7NTD.

**Figure supplement 5.** Model of Sld3CTD on the Cdc45–MCM–GINS (CMG).

(Sld3CBD). Sequence alignment of Sld3CBD among Sld3 and Treslin revealed that all Cdc45-binding residues in α8 and α9 identified in our study were almost conserved or exhibited conserved changes (*Figure 1—figure supplements 5 and 6*). This conservation suggests that these regions provide a similar interaction manner between Sld3CBD and Cdc45 in the regulation of metazoan DNA replication. Therefore, we hypothesize that Treslin may load Cdc45 as observed in yeast Sld3 and Sld7.

By structural comparison, we found that Sld3CBD and Cdc45 changed their conformations to bind to each other. The conformational changes in Cdc45 DHHA1 upon binding to Sld3CBD also caused the contact between Cdc45 and Mcm2NTD in the Sld3CBD–Cdc45–MCM–dsDNA and SCMG–dsDNA models, whereas DHHA1 interacted with neither MCM nor GINS in the CMG structure. Taking the structural information together, Sld3 seems to play a guiding role in helping Cdc45 bind to MCM at the right position. Furthermore, Sld3CBDs in each monomer of the Sld3CBD–Cdc45–MCM dimer were located at a distance of more than 230 Å, which is consistent with the results of a particle size analysis of the Sld7–Sld3ΔC–Cdc45 complex off MCM DH in solution. Therefore, we propose that a binding manner of Sld7–Sld3–Cdc45 in a flexible long-shaped dimer Cdc45–Sld3–(Sld7)$_2$–Sld3–Cdc45 off/on MCM DH is advantageous for efficiently recruiting two Cdc45 molecules to an MCM DH, consequently leading to the formation of a pair of CMG helicases.

In the SCMG–dsDNA complex model, Cdc45 bound to Sld3CBD, MCM, and GINS on different sides (with contact surfaces of 6.7, 4.8, and 5.1% of the total Cdc45 surface, respectively). Our structure of Sld3CBD-Cdc45 and models show that these bindings occur at distinct sites on Cdc45, suggesting that Sld3CBD, Cdc45, and GINS could bind to MCM together without steric clash. The competition between Sld3 and GINS for binding to Cdc45 or Cdc45-MCM (by mixing them in vitro) reported by Bruck et al. (*Bruck and Kaplan, 2011*) may be caused by the conformational change of Cdc45 DHHA1 or the lack of other auxiliary initiation factors, indicating that activated CMG formation requires regulation. In particular, Sld3 and GINS bind to opposite positions of Cdc45 and MCM ring (Mcm2–4–6 vs Sld3 and Mcm5–3–7 vs GINS), suggesting that the GINS-recruitment protein should cross a long distance in an MCM monomer or MCM DH to access the phosphorylation site (T600 and S622) of Sld3. Furthermore, our SCMG–dsDNA model revealed that Sld3CBD on CMG appears to contact an N-terminal helix of Mcm2CTD, while Sld3CTD may extend to bind to Mcm4NTD through interaction with the Mcm2NTD and Mcm6NTD (*Figure 4—figure supplement 5*). These findings suggest that the Sld3–Sld7 binding to MCM does not interfere with the AAA+ motors' ability to regulate MCM ring dynamics during its opening/closing via the gap between Mcm2CTD and Mcm5CTD (*Abid Ali et al., 2017*; *Yuan et al., 2016*; *Noguchi et al., 2017*; *Li et al., 2015*). Taking our findings and those of previous studies together, we propose a detailed process for helicase CMG formation from inactive MCM, as depicted in *Figure 5A–C*: Sld7–Sld3 brings Cdc45 onto MCM as a Sld7–Sld3–Cdc45 dimer (Cdc45–Sld3–[Sld7]$_2$–Sld3–Cdc45), and remains until GINS loading.

The following inquiry concerns the dissociation of Sld3 and other factors. Interestingly, the mutant analysis demonstrates that disrupting a single binding site between Sld3CBD and Cdc45 suffices to dissociate Sld3CBD and Cdc45, indicating that a functionally critical binding between Sld3CBD and Cdc45 can be broken easily. Furthermore, our binding analysis of ssARS1 fragments to Sld3CBD, Sld3CBD–Cdc45, Sld7–Sld3ΔC–Cdc45 and Sld7–Sld3ΔC showed the sequence specificity to ssARS1-2 and ssARS1-5 fragments, not others. Considering that ssARS1-2 and ssARS1-5 are on both sides of MCM DH-bound dsDNA at replication origin (*Chang et al., 2011*), the origin unwinding by CMG generates ssDNA and further sequesters the Sld7-Sld3 complex onto ssDNA to remove Sld7-Sld3 from CMG. As a bridge protein, Sld3 recruits Cdc45 to MCM, and its next phosphorylated state regulates the subsequent recruitment of GINS loading with Dpb11–Sld2 (*Muramatsu et al., 2010*). Thus, the release of Sld3 and Sld7 from CMG could be associated with unwound ssARS1 and may also be related to the dissociation of Dpb11–Sld2 from CMG (*Figure 5D and E*; *Lewis et al., 2022*; *Douglas et al., 2018*). Furthermore, our proposals require a visualization of the Sld3–Sld7–Cdc45–MCM complex structure during GINS recruitment to establish the complete CMG formation process.

In conclusion, our structural and biochemical studies of Sld3CBD–Cdc45 revealed a detailed process of CMG formation and the subsequent ssDNA-mediated release of the central regulator Sld3 with other factors, leading to a deeper understanding of the initiation mechanism of DNA replication.

# Materials and methods

## Preparation of proteins

The C-terminal His-tagged Sld3CBD of *S. cerevisiae* was expressed in *Escherichia coli* and prepared as previously described (*Itou et al., 2014*). For co-expression of *S. cerevisiae* Sld3CBD (S148–K430) and Cdc45 (M1-L650) (Sld3CBD–Cdc45) in *E. coli*, the Sld3CBD DNA fragment containing the His$_6$-tag (LEHHHHHH) at the C-terminus and the Cdc45 DNA fragment were cloned in the co-overexpression vector pETDuet-1 between the NcoI and SalI restriction sites and the NdeI and XhoI restriction sites, respectively. The primer sequences for Sld3CBD–Cdc45 are listed in *Supplementary file 1*.

To confirm the interactions obtained from the Sld3CBD–Cdc45 structure, we constructed four types of single or multi-mutants for Sld3CBD (Sld3-3S: I352S/I355S/L356S, Sld3-3E: I352E/I355E/L356E, Sld3-2R: D344R/D348R, and Sld3-Y: I352Y), and three types of single or multi-mutants for Cdc45 (Cdc45-IIIS: L522S/L527S/V529S, Cdc45-IIIE: L522E/L527E/V529E, Cdc45-IIS: L637S/L641S, Cdc45-IIE: L637E/L641E, and Cdc45-A: R523A) using the Quick Change site-directed mutagenesis method and the inverse PCR method with pETDuet-1–Sld3CBD–Cdc45 as template DNA. The primer sequences for the mutant strains are listed in *Supplementary file 1*.

As *S. cerevisiae* Sld7–Sld3–Cdc45 could not be co-overexpressed in *E. coli*, we attempted to clone it from several other fungal sources. As a result, complex of Sld7 (M1-T268), Sld3ΔC (M1-K430, truncated C-terminal domain), and Cdc45 (M1-I666) from the budding yeast *Kluyveromyces marxianus*, which belongs to the same family as *S. cerevisiae* (Sld7–Sld3ΔC–Cdc45) was obtained. Sld7, Sld3 and Cdc45 have sequence conservation with similar structures (RMSD = 0.356, 1.392, and 0.891 for Cα atoms of Sld7CTD, Sld7NTD-Sld3NTD, and Sld3CBD-Cdc45) predicted by the Alphafold3 (*Abramson et al., 2024*). Sld7–Sld3ΔC–Cdc45 was cloned into pETDuet-1 with a His$_6$-tag (LEHHHHHH) at the C-terminus of Sld3ΔC. The primer sequences for Sld7–Sld3ΔC and Sld7–Sld3ΔC–Cdc45 are listed in

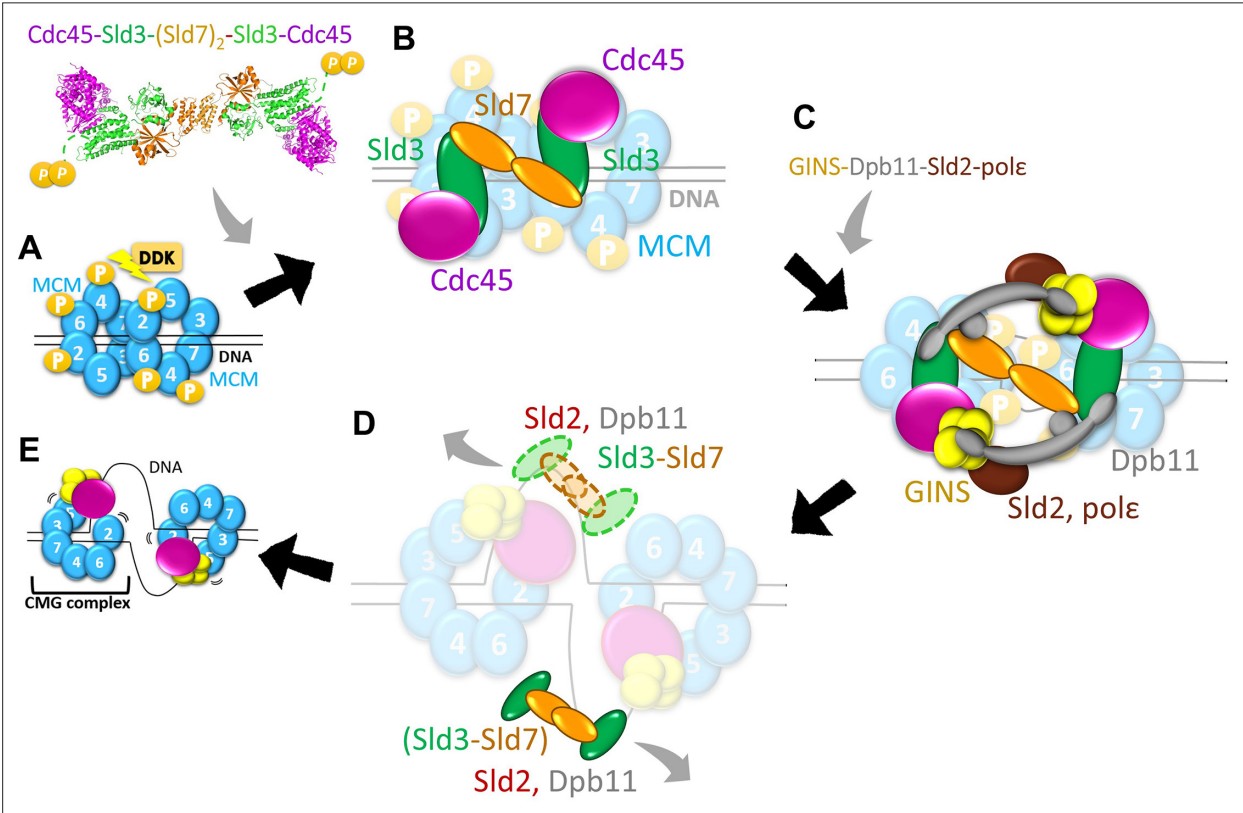

**Figure 5.** Proposal for Cdc45–MCM–GINS (CMG) formation with Sld7–Sld3. (**A**) Phosphorylation of Mcm2,4,6 by DDK after MCM double hexamer (DH) was loaded on double-stranded DNA (dsDNA) at the replication origin. (**B**) Cdc45 recruitment to MCM DH by Cdc45–Sld3–[Sld7]$_2$–Sld3–Cdc45. (**C**) After CDK-mediated phosphorylation of Sld3CTD in Cdc45–MCM, Dpb11–Sld2 recruits GINS and polε to Sld7–Sld3-Cdc45–MCM to form an active helicase CMG. (**D**) Unwinding of dsDNA by CMG with MCM DH separation and MCM ring opening. Sld3 and other factors are released upon binding to single-stranded DNA (ssDNA). (**E**) Each CMG unwinds the dsDNA in two directions, initiating DNA replication.

*Supplementary file 1*. As Cdc45 mutants can lose the ability to bind to Sld3, we overexpress Sld7–Sld3ΔC by using multi-mutants of Cdc45 (Cdc45-IIS) in Sld7–Sld3ΔC–Cdc45. The Ser-substitution residues (L654S/L658S) of Cdc45-IIS from *K. marxianus* were selected based on sequence conservation. We constructed mutant Sld7–Sld3ΔC–Cdc45-IIS using the Quick Change site-directed mutagenesis method and the inverse PCR method with pETDuet-1–Sld7–Sld3ΔC–Cdc45 as the template DNA. The primer sequences for the mutant strains are listed in *Supplementary file 1*.

To overexpress Sld3CBD, Sld3CBD–Cdc45, Sld3CBD–Cdc45 mutants, Sld7–Sld3ΔC–Cdc45 and Sld7–Sld3ΔC, the vector of pET26Sld3CBD, pETDuet-1-Sld3CBD–Cdc45, pETDuet-1-Sld3CBD–Cdc45 mutants, pETDuet-1-Sld7–Sld3ΔC–Cdc45 or pETDuet-1-Sld7–Sld3ΔC was transformed into *E. coli* strain BL21 (DE3) through electroporation, followed by preculturing in 5 ml of Luria–Bertani medium containing 100 µg/mL ampicillin at 310 K overnight. The culture was then transferred to 3 L Luria–Bertani medium and grown until the $OD_{600}$ reached 0.6. After cooling the culture for 30 min on ice, overexpression of each sample was induced by the addition of isopropyl-β-D-1-thiogalactopyranoside to a final concentration of 0.5 mM, and then cells were grown for an additional 12 hr at 293 K. Cells were harvested by centrifugation at 4000 *g* for 20 min at 283 K and then resuspended in a buffer containing 20 mM Tris-HCl pH 7.5, 300 mM NaCl, 10% glycerol, 0.1 mg/ml DNase, and 1× protease-inhibitor cocktail (cOmplete EDTA-free; Roche, Basel, Switzerland).

The harvested cells were crushed through sonication and centrifuged at 40,000 *g* for 30 min at 283 K. After filtration through a 0.45 µm filter (Sigma-Aldrich/Merck Millipore, Burlington, MA, USA), the supernatant was then loaded onto the HisTrap HP column equilibrated with buffer A [20 mM Tris-HCl, pH 7.5, 300 mM NaCl, 10% glycerol] and the column was washed with buffer A. The His-tagged target protein was eluted with imidazole at 20% (100 mM), 30% (150 mM), and 50% (250 mM), followed by a linear gradient of 250–500 mM imidazole in buffer B [20 mM Tris-HCl, pH 7.5, 300 mM NaCl, 10% glycerol, 500 mM imidazole]. We analyzed the fractions using sodium dodecyl sulfate-polyacrylamide gel electrophoresis (SDS-PAGE). Except for mutants of Sld3CBD–Cdc45, the pooled fractions were then purified through size-exclusion chromatography with a buffer A-equilibrated Superdex 200 16/60 column (GE HealthCare, Chicago, IL, USA). Protein purity was confirmed through SDS-PAGE. The purified samples were concentrated to 10 mg/ml and stored at 193 K.

## Crystallization and diffraction data collection

The crystallization screening of Sld3CBD–Cdc45 was performed at 293 K using the sitting-drop vapor diffusion method. The drop was a mixture of 1.0 µl of a 10 mg/ml protein solution with the equivalent volume of reservoir buffer from the commercial crystallization kits (JCSG core I-IV, classics, classics II, PEGs, PEGsII, and MPD suite from Qiagen, Venlo, the Netherlands). The initial needle- and plate-shaped crystals appeared under eight and one conditions, respectively. After optimizing the conditions by altering the type and concentration of precipitant, the salt reagent, and the pH of the buffer, the best crystals were obtained in a solution of 0.2 M sodium acetate, 0.1 M Bis-Tris propane (pH 6.5: pH 8.5=3:7), 20%(w/v) PEG3500, and a drop containing 2.0 µl of a 10 mg/ml protein solution mixed with the equivalent volume of reservoir buffer.

Diffraction data were collected using a beamline BL-17A at the Photon Factory, Tsukuba, Japan. Before data collection, the crystals were cryoprotected by soaking them in a reservoir buffer supplemented with 5% (v/v) glycerol, and then flash-cooled under a nitrogen gas stream at 100 K. The *XDS/XSCALE* program was used to index, integrate, scale, and merge the dataset (*Kabsch, 2010*). *Supplementary file 2* summarizes the statistics of data collection and processing.

## Structure determination and refinement

The structure of Sld3CBD–Cdc45 was determined using the molecular replacement method with the *AutoMR* wizard in *Phenix* (*McCoy et al., 2007*; *Adams et al., 2010*). The structures of human Cdc45 (PDBID: 5DGO) (*Simon et al., 2016*) and Sld3CBD (PDBID: 3WI3) (*Itou et al., 2014*) were used as search models. Several rounds of refinement were performed using the *Phenix_refine* program in *Phenix*, interleaved with manual building and fitting according to the electron density maps of 2Fo-Fc and Fo-Fc using the *Coot* program (*Adams et al., 2010*; *Emsley and Cowtan, 2004*). *Supplementary file 2* presents the final refinement statistics and geometry defined by MolProbity (*Chen et al., 2010*). All structural diagrams were generated using *PyMol* (*Delano, 2002*).

## Mutant analysis of Sld3 and Cdc45

To analyze the binding sites of Sld3CBD-Cdc45, in conjunction with Cdc45 binding sites to MCM and GINS, we performed a co-express pull-down binding assay. We constructed four variants of Sld3CBD and five variants of Cdc45 according to the binding information from our Sld3CBD–Cdc45 structure. We co-express all mutations of Sld3CBD-Cdc45 (Sld3CBD-C-histag) and load them onto the HisTrap HP column under the same conditions as in [Preparation of proteins]. After extracting the samples through Ni-affinity chromatography, we concentrated each eluted sample to 0.005 mg/mL (by nano-drop A280) and checked the binding status of the mutants using SDS-PAGE. Both Sld3CBD and Cdc45 should be observed in the elution group if they form a complex. The overexpressed level of the Cdc45 was checked by -IPTG and +IPTG.

We performed circular dichroism (CD) spectrometry measurement of Sld3 mutants–Cdc45 to check whether mutations affected the structure. CD spectra were collected using a J-805 spectropolarimeter (JASCO, Tokyo, Japan) in a quartz cell with a path length of 1 mm in an atmosphere of $N_2$ at 298 K. For CD measurements, the samples were dialyzed in a buffer [20 mM Tris-HCl, pH 7.5, 50 mM NaCl] and adjusted to a concentration of 0.5 mg/mL through absorption. CD spectra for the wavelength range of 190–300 nm were obtained by averaging the results of four scans. The results are given in molar ellipticity per residue $[\theta]$ mrw ($\times 10^{-3}$) vs wavelength/nm (*Greenfield, 2006*). The secondary structures of each sample were estimated using the K2D3 method (*Louis-Jeune et al., 2012*). For wild-type Sld3CBD, we calculated secondary structures from the obtained Sld3CBD–Cdc45 structure.

## Growth of mutant cells

The isolation and plasmid shuffling of the temperature-sensitive yeast strain YYK13 (for Sld3 mutant analysis) and Cdc45-35 (for Cdc45 mutant analysis) have been described in a previous study (*Kamimura et al., 2001*). To investigate Sld3 mutants, strain YYK13 was transformed with the YCplac22 plasmid containing multiple variants of *SLD3* mutant genes. The transformants were streaked on SD plates lacking Trp and Leu (SD-Trp, Leu) and SD-Trp, Leu containing 5-fluoroorotic acid (5-FOA-Trp, Leu), and then incubated at 298 K for 3 days. To analyze the Cdc45 mutant, strain Cdc45-35 containing the *CDC45* mutant gene was re-transformed with the YEplac195 plasmid containing *SLD3* and the YEplac122 plasmid containing *SLD7*. The transformants were streaked on yeast extract–peptone–dextrose plates and incubated at 298 or 305 K for 3 days.

## Modelling of complexes

We constructed the models of Sld3CBD–Cdc45–MCM–dsDNA (Sld3CBD–Cdc45–MCM dimer complexed with dsDNA) and SCMG–dsDNA (SCMG dimer complexed with dsDNA) using a two-step process. First, the structure of the MCM–NTDs in the CMG monomer (PDBID: 3JC6) (*Yuan et al., 2016*) was superimposed on that of MCM DH complexed with dsDNA (5BK4) (*Noguchi et al., 2017*) to obtain a dimer of Cdc45–MCM–GINS complexed with dsDNA (Cdc45–MCM–GINS–dsDNA). Next, the models of Sld3CBD–Cdc45–MCM–dsDNA and SCMG–dsDNA were obtained by superimposing Cdc45NTD from Sld3CBD–Cdc45 onto Cdc45–MCM–GINS–dsDNA.

## Dynamic light scattering

We estimated the particle size of the Sld7–Sld3ΔC–Cdc45 complex using dynamic light scattering in terms of peak size (*Stetefeld et al., 2016*). To avoid the effect of concentration, we measured four samples of the Sld7–Sld3ΔC–Cdc45 complex. Each sample was overexpressed independently and purified through size-exclusion chromatography. All samples were concentrated to 10 mg/ml, and stored at 193 K. Before measurement, protein samples were dialyzed in a buffer containing 20 mM Tris-HCl pH 7.5 and 300 mM NaCl and then filtered through a 0.22 μm filter. The pure buffer was measured as a background control and 10 μM lysozyme was measured as a standard control. Dynamic light scattering data were collected and analyzed using a Malvern Zetasizer Nano-ZS instrument (Malvern Panalytical, Malvern, UK) for 0.2–0.5 mg/ml, 500 μl protein samples in a microquartz cuvette (Malvern-ZEN0112). An automatic duration model was used to collect data. The Zetasizer 6.20 was utilized for data analysis from three measurements to estimate the particle size of each sample.

## Electrophoretic mobility shift assay for ssDNA binding

Sld3 binds to single strands of DNA (*ssARS1-2* and *ssARS1-5*) (*Bruck and Kaplan, 2011*). To determine the specificity of the ssDNA binding affinity of Sld3, we conducted an electrophoretic mobility shift assay (EMSA) using non-denaturing PAGE (native-PAGE) (*Hellman and Fried, 2007*). Given that Cdc45 binds ssDNA with a nonspecific sequence at lengths greater than 60 bases (*Bruck and Kaplan, 2013*), we designed three fragments of ssDNA 40 bases in length (first half 1–40 bp, second half 41–80 bp, and middle half 21–60 bp) for each ssDNA 80 bp segment. All ssDNA fragments of *S. cerevisiae* were synthesized by Sigma-Aldrich. The 40 bp dsDNA fragments (dsARS1-34_1:ssARS1-3_1/ssARS1-4_2) were converted by annealing them using the PCR protocol and then checked through polyacrylamide gel electrophoresis. The loaded samples were incubated overnight at 277 K in TMK buffer (20 mM Tris-HCl pH8, 100 mM $MgCl_2$, 200 mM KCl) containing synthesized ssDNA and varying amounts of proteins at ssDNA: protein molecular ratios of 1:0, 1:0.5, 1:1, 1:2, and 0:1 with 20 pM as 1 unit. After incubation, the mixtures were loaded onto a polyacrylamide gel (5% (w/v) acrylamide (39:1), 10% 10 ×running buffer (0.25 M Tris, 1.92 M Glycine), 0.1% Ammonium peroxodisulphate, 0.06% (v/v) TEMED) without denaturing (native-PAGE). EMSA was performed at 10 mA/200 V per gel for 40 min at 277 K in 1× running buffer. After electrophoresis, the reaction products were visualized using Fast Blast DNA stain (Bio-Rad Laboratories, Hercules, CA, USA) (100 × Fast Blast DNA stain diluted by 1× running buffer) or SYBR safe (Sigma-Aldrich, Burlington, MA, USA) (SYBR safe:1 × running buffer = 0.0001:1) to stain the ssDNA and Coomassie Brilliant Blue (G-250) to stain the proteins. We repeated the EMSA experiments three or more times to confirm the readability. All EMSA results were converted into 8-bit images after brightness and contrast normalized, then the background was removed, and the integrated grayscale was calculated using *ImageJ* (*Abràmoff et al., 2004*). The results of band-integrated grayscale calculation were performed with a t-test and plotted using *GraphPAD Prism* (Graphpad Software, San Diego, CA, USA). Considering the functional similarity of ARS1-core, the EMSA of Sld7–Sld3ΔC and Sld7–Sld3ΔC–Cdc45 of *K. marxianus* used ssDNA fragments of *S. cerevisiae*.

## Acknowledgements

The authors thank Mr. Naofumi Sakurai for his help in the protein purifications. We would like to thank Dr. Nobutaka Shimizu and Dr. Kento Yonezawa for SEC-SAXS experiments, and the beamline staff of the Photon Factory and SPring-8 for collecting X-ray diffraction data (Proposal No. 2016A2562, 2017 A2551, and 2018 A2508)

## Additional information

### Funding

| Funder | Grant reference number | Author |
|---|---|---|
| Japan Society for the Promotion of Science, KAKENHI Grant-in-aid for Scientific Research(B) | 21H01754 | Min Yao |
| Japan Agency for Medical Research and Development, Platform Project for Supporting Drug Discovery and Life Science Research | JP18am0101071 | Min Yao |
| Japan Agency for Medical Research and Development, Platform Project for Supporting Drug Discovery and Life Science Research | JP19am0101083 | Min Yao |

| Funder | Grant reference number | Author |
|---|---|---|

The funders had no role in study design, data collection and interpretation, or the decision to submit the work for publication.

## Author contributions

Hao Li, Formal analysis, Investigation, Writing – original draft; Izumi Ishizaki, Formal analysis, Investigation; Koji Kato, Toyoyuki Ose, Formal analysis; Xiaomei Sun, Sachiko Muramatsu, Investigation; Hiroshi Itou, Investigation, Writing – review and editing; Hiroyuki Araki, Conceptualization, Formal analysis, Writing – review and editing; Min Yao, Conceptualization, Formal analysis, Methodology, Writing – review and editing

## Author ORCIDs

Hao Li (ID) https://orcid.org/0000-0001-7885-9841
Hiroyuki Araki (ID) https://orcid.org/0000-0001-5405-0677
Min Yao (ID) https://orcid.org/0000-0003-1687-5904

Reviewer #1 (Public review): https://doi.org/10.7554/eLife.101717.4.sa1
Reviewer #2 (Public review): https://doi.org/10.7554/eLife.101717.4.sa2
Reviewer #3 (Public review): https://doi.org/10.7554/eLife.101717.4.sa3
Author response https://doi.org/10.7554/eLife.101717.4.sa4

# Additional files

## Supplementary files

Supplementary file 1. Primers used in this study.

Supplementary file 2. Statistics of data collection and refinement.

MDAR checklist

## Data availability

All data generated or analysed during this study are included in the manuscript and supporting files. Diffraction data and structure have been deposited in PDB under the accession code 8J09.

The following dataset was generated:

| Author(s) | Year | Dataset title | Dataset URL | Database and Identifier |
|---|---|---|---|---|
| Li H, Ishizaka I, Kato K, Sun X, Muramatsu S, Itou H, Ose T, Araki H, Yao M | 2023 | Crystal structure of the Sld3 Cdc45-binding-domain, in complex with Cdc45 | https://doi.org/10.2210/pdb8J09/pdb | Worldwide Protein Data Bank, 10.2210/pdb8J09/pdb |

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
