## [Editor Report · eLife Assessment]

This **valuable** paper describes the crystal structure of a complex of the Sld3-Cdc45-binding domain (CBD) with Cdc45, which is essential for the assembly of an active Cdc45-MCM-GINS (CMG) double-hexamer at the replication origin. The structural and biochemical analyses of protein-protein interactions and DNA binding provided **solid** evidence to support the authors' conclusion. The results shown in the paper are of interest to researchers in DNA replication and genome stability.

---

## [Referee Report · Reviewer #1 (Public review)]

Summary:

The crystal structure of the Sld3CBD-Cdc45 complex presented by Li et al. is a significant contribution that enhances our understanding of CMG formation during the rate-limiting step of DNA replication initiation. This structure provides crucial insights into the intermediate steps of CMG formation, and the particle analysis and model predictions compellingly describe the mechanism of Cdc45 loading.

Building upon previously known Sld3 and Cdc45 structures, this study offers new perspectives on how Cdc45 is recruited to MCM DH through the Sld3-Sld7 complex. The most notable finding is the structural rearrangement of Sld3CBD upon Cdc45 binding, particularly the α8-helix conformation, which is essential for Cdc45 interaction and may also be relevant to its metazoan counterpart, Treslin. Additionally, the conformational shift in the DHHA1 domain of Cdc45 suggests a potential mechanism for its binding to Mcm2NTD.

Furthermore, the ssDNA-binding experiments involving Sld3 further support a broader functional role in the replication process, beyond its established role in recruiting Cdc45. This adds an intriguing new layer to our understanding of Sld3's activity in the yeast.

---

## [Referee Report · Reviewer #2 (Public review)]

Summary

The manuscript presents valuable findings, particularly in the crystal structure of the Sld3CBD-Cdc45 interaction and the identification of additional sequences involved in their binding. The modeling of the Sld7-Sld3CBD-CDC45 subcomplex is novel, and the results provide insights into potential conformational changes that occur upon interaction. Although the single-stranded DNA binding data from Sld3 of different species is a minor weakness, the experiments support a model in which the release of Sld3 from the complex may be promoted by its binding to origin single-stranded DNA exposed by the helicase.

---

## [Referee Report · Reviewer #3 (Public review)]

Summary:

The paper by Li et al. describes the crystal structure of a complex of Sld3-Cdc45-binding domain (CBD) with Cdc45 and a model of the dimer of an Sld3-binding protein, Sld7, with two Sld3-CBD-Cdc45 for the tethering. In addition, the authors showed the genetic analysis of the amino acid substitution of residues of Sld3 in the interface with Cdc45 and biochemical analysis of the protein interaction between Sld3 and Cdc45 as well as DNA binding activity of Sld3 to the single-strand DNAs of the ARS sequence.

---

## [Author Response]

The following is the authors’ response to the previous reviews

**Reviewer #1 (Public review):**
Summary:The crystal structure of the Sld3CBD-Cdc45 complex presented by Li et al. is a significant contribution that enhances our understanding of CMG formation during the rate-limiting step of DNA replication initiation. This structure provides crucial insights into the intermediate steps of CMG formation, and the particle analysis and model predictions compellingly describe the mechanism of Cdc45 loading. Building upon previously known Sld3 and Cdc45 structures, this study offers new perspectives on how Cdc45 is recruited to MCM DH through the Sld3-Sld7 complex. The most notable finding is the structural rearrangement of Sld3CBD upon Cdc45 binding, particularly the α8-helix conformation, which is essential for Cdc45 interaction and may also be relevant to its metazoan counterpart, Treslin. Additionally, the conformational shift in the DHHA1 domain of Cdc45 suggests a potential mechanism for its binding to Mcm2NTD. Furthermore, Sld3's ssDNA-binding experiments provide evidence of its novel functions in the DNA replication process in yeast, expanding our understanding of its role beyond Cdc45 recruitment.Strengths:The manuscript is generally well-written, with a precise structural analysis and a solid methodological section that will significantly advance future studies in the field. The predictions based on structural alignments are intriguing and provide a new direction for exploring CMG formation, potentially shaping the future of DNA replication research. This research also opens up several new opportunities to utilize structural biology to unravel the molecular details of the model presented in the paper.Weaknesses:The main weakness of the manuscript lies in the lack of detailed structural validation for the proposed Sld3-Sld7-Cdc45 model, and its CMG bound models, which could be done in the future using advanced structural biology techniques such as single particle cryo-electron microscopy. It would also be interesting to explore how Sld7 interacts with the MCM helicase, and this would help to build a detailed long-flexible model of Sld3-Sld7-Cdc45 binding to MCM DH and to show where Sld7 will lie on the structure. This will help us to understand how Sld7 functions in the complex. Also, future experiments would be needed to understand the molecular details of how Sld3 and Sld7 release from CMG is associated with ssARS1 binding.

The proposals based on this study provide new knowledge of the CMG formation process. We agree that our Sld3-Sld7-Cdc45 model will be further confirmed by cryo-EM. We improved our ssARS1-binding assay and quantified data (See the response to Recommendations for the authors of #3 review).

**Reviewer #2 (Public review):**
SummaryThe manuscript presents valuable findings, particularly in the crystal structure of the Sld3CBD-Cdc45 interaction and the identification of additional sequences involved in their binding. The modeling of the Sld7-Sld3CBD-CDC45 subcomplex is novel, and the results provide insights into potential conformational changes that occur upon interaction. Although the single-stranded DNA binding data from Sld3 of different species is a minor weakness, the experiments support a model in which the release of Sld3 from the complex may be promoted by its binding to origin single-stranded DNA exposed by the helicase.StrengthsThe Sld3CBD-Cdc45 structure is a novel contribution, revealing critical residues involved in the interaction.The model structures generated from the crystal data are well presented and provide valuable insights into the interaction sequences between Sld3 and Cdc45.The experiments testing the requirements for interaction sequences are thorough and conducted well, with clear figures supporting the conclusions.The conformational changes observed in Sld3 and Cdc45 upon binding are interesting and enhance our understanding of the interaction.The modeling of the Sld7-Sld3CBD-CDC45 subcomplex is a new and valuable addition to the field.The proposed model of Sld3 release from the complex through binding to single stranded DNA at the origin is intriguing.WeaknessesThe section on the binding of Sld3 complexes to origin single-stranded DNA is somewhat weakened by the use of Sld3 proteins from different species. The comparisons between Sld3-CBD, Sld3CBD-Cdc45, and Sld7-Sld3CBD-Cdc45 involve complexes from different species, limiting the comparisons' value.Although the study reveals that Sld3 binds to different residues of Cdc45 than those previously shown to bind Mcm or GINS, the data in the paper do not shed any additional light on how GINS and Sld3 binding to Cdc45 or Mcms. would affect each other. Other previous research has suggested that the binding of GINS and Sld3 to Mcm or Cdc45 may be mutually exclusive. The authors acknowledge that a structural investigation of Sld3, Sld7, Cdc45, and MCM during the stage of GINS recruitment will be a significant goal for future research.

We agree that it is better to use all samples from a source; however, due to limitations in protein expression, we used Sld7-Sld3CBD-Cdc45 from a different source. The two sources used in this study belong to the same family, and the proteins Sld7, Sld3 and Cdc45 share sequence conservation with similar structures predicted by Alphafold3 (RMSD = 0.356, 1.392, and 0.891 for Ca atoms of Sld7CTD, Sld7NTD-Sld3NTD, and Sld3CBD-Cdc45). Such similarity in source and proteins allows us to do the comparison. We also mentioned that a cryo-EM study of Sld3-Sld7-Cdc45-MCM and Sld3-Sld7-CMG structures will be a significant goal for future research in our manuscript.

**Reviewer #3 (Public review):**
Summary:The paper by Li et al. describes the crystal structure of a complex of Sld3-Cdc45-binding domain (CBD) with Cdc45 and a model of the dimer of an Sld3-binding protein, Sld7, with two Sld3-CBD-Cdc45 for the tethering. In addition, the authors showed the genetic analysis of the amino acid substitution of residues of Sld3 in the interface with Cdc45 and biochemical analysis of the protein interaction between Sld3 and Cdc45 as well as DNA binding activity of Sld3 to the single-strand DNAs of the ARS sequence.Strengths:The authors provided a nice model of an intermediate step in the assembly of an active Cdc45-MCM-GINS (CMG) double hexamers at the replication origin, which is mediated by the Sld3-Sld7 complex. The dimer of the Sld3-Sld7 complexes tethers two MCM hexamers together for the recruitment of GINS-Pol epsilon on the replication origin.Weaknesses:The biochemical analysis should be carefully evaluated with more quantitative ways to strengthen the authors' conclusion even in the revised version.

In this revision, we improved our ssARS1-binding assay in more quantitative ways (See the response to Recommendations for the authors).

**Reviewer #1 (Recommendations for the authors):**
I thank the authors for all their replies to my previous questions and for doing all the necessary corrections. I am satisfied with most of their replies, however, upon second reading I have a few more suggestions which could help to improve the manuscript further and make an impact in the field. My comments are listed below.(1) In general, the manuscript is well structured, but I feel that it requires professional English correction. In many places it was difficult to understand the sentences and I had to read it several times to understand it. Also, very long sentences should be avoided. The flow should be easy to read and understand, and that is why I feel it requires professional English correction.

Following the comment, we checked English carefully and shortened the very long sentences.

(2) Page 5, line 103, please include molecule after the word complex to make it like- "Only one complex molecule exists within an asymmetric unit."

We revised this sentence (P5/L103).

(3) Line 113- more than the N-terminal half of the protruding long helix α7 113 was disordered in the Sld3CBD-Cdc45 complex. This sentence is not clear. What does it mean more than the N-terminal half? Please rewrite it.

We revised this sentence to give the corresponding residue number “(D219–H231)” (P5/L114).

(4) Page 5, result 2- Conformation changes in Sld3CBD and Cdc45 for binding each other, this section may require a little restructuring. Line 130-131- "Therefore, the helix α8CTP seems to be an intrinsically disordered segment when Sld3 alone but 130 folds into a helix coupled to the binding partner Cdc45 in the Sld3CBD-Cdc45 complex." This statement is the crux of the structural finding and therefore, I feel it should move after the first sentence.

Thank you for your comments. We rewrote this part (P5/L128-131).

(5) Line 121-122: Compared to the isolated form (PDBIDs: 5DGO 121 for huCdc45 [31] and 6CC2 for EhCdc45 [33]) and the CMG form (PDBID: 3JC6). Write it in the same format. Make 6CC2 in bracket like other PDB IDs. Restructure this sentence.

We revised this sentence (P5/122-123).

(6) Line 127-129: This sentence is also not very clear.

We revised this sentence together with above No (4). (P5/L128-131)

(7) In my question 4- "Can authors add a supplementary figure showing the probability of disordernes..."., I meant to use a disorder prediction tool like IUPred for the protein sequences and show that α8 is predicted to be a disordered upon sequence analysis. This will help to show the inherent property of α8 helix, and it could add up to the understanding that a disordered region is being structured in the complex structure.

The structures showed that α8CTP is stabilized by binding with Cdc45, but disordered in Sld3CBD alone, indicating that this part is flexible, like an intrinsically disordered segment. We have deposited the structure to PDB, so predictions like IUPred cannot show meaningful information.

(8) Question 9 regarding Supplementary Figure 8- Please include your statement in the figure legend - "WT Sld3CBD was prepared in a complex with Cdc45, while the mutants of Sld3CBD existed alone, we calculated the elements of secondary structure from the crystal structure of Sld3CBD-Cdc45. The concentration of samples was controlled to the same level for CD measurement."

Following the comment, we optimized the figure legend of Supplementary Figure 8.

(9) Question 13- I understand that negative staining and SEC-SAXS experiments could be very tricky for such protein complexes, which have very long loops and are flexible. Did authors try a GraFix cross-linking before doing the negative staining TEM? If it is not being tried, then it might be a good idea to try it and it may help to get much cleaner particles and easier class averaging. Although I completely understand the technical challenges the authors describe and I agree with them, I still feel that one good experiment that shows this dimer model would be very helpful to strengthen the claim. I am concerned because if people start using a similar DLS experiment to calculate intermolecular distances, citing your paper, in many cases it might be a wrong interpretation. In case the negative staining still does not work, at least discuss your technical challenges in the discussion section and mention that SEC-SAXS showed a similar length of the complex and show the Guinier plot and Porod plots in the supplementary data.

We believe that DLS is one of the methods for analyzing the single particle size. Of course, the confirmation by multiple methods will give compelling evidence. Following the comment, we added SEC-SAXS data in the [Results] (P7/L194-196) (Cdc45 recruitment to MCM DH by Sld3 with partner Sld7) and Supplementary Figure 11. The Sld7-Sld3-Cdc45 forms a flexible, long shape. Each binding domain is rigid but linked by the long loops. The flexibility problems are caused by the long loop linkers, but not by binding. So, we did not try to use the cross-linking method for analysis experiments.

(10) Page 8, line 221- litter sequence specificity: Correct the word "litter" with little. Also, the word shaped is written as sharped at a few places in the manuscript. Please correct it.

We apologize for making such mistakes. We have modified these words.

(11) Page 9, line 237-238: Would it be possible to add a lane showing Sld7 binding to the ssDNA in figure 4. I recommend showing this to understand the ssDNA binding affinity of Sld7 by itself and it will also help us to compare when it is in complex with Sld3.

Considering that Sld7 on CMG is always a complex with Sld3, the ssDNA binding affinity should use the Sld3-Sld7 complex. Additionally, we attempted to overexpress Sld7, but could not obtain the target protein.

**Reviewer #2 (Recommendations for the authors):**
Thank you for the improved manuscript. The following sentence is unclear: "Cdc45 binds tighter to long ssDNA (>60 bases) with a litter sequence specificity".

We apologize for making such a mistake. We modified “litter” to “little”.

I found it challenging to understand which species were used while reading the results section and figure legends. I recommend that the authors revise the text in both the results and figure legends to clearly indicate when proteins from different species are being compared. Additionally, it would be valuable to explicitly acknowledge this limitation in the text.

Following the comment, we added a description for using different species in results (P8/L224-225) and figure legends (Supplementary Figure 14). We added more information in the Methods to explain why we used two species for preparing proteins.

**Reviewer #3 (Recommendations for the authors):**
Major points:(1) The current title is not appropriate for the general readers. At least, DNA replication or DNA replication initiation should be added and abbreviations such as CBD should be avoided.

Following the comment, we added “DNA replication” into the title. Regarding “CBD”, since the full name of “Cdc45 binding domain” is too long, we continue to use Sld3CBD.

(2) As in my previous review, I asked for quantification of the EMSA assay shown in Figure 4 and Supplemental Figures 13 and 14. Since some signals of the bands are very weak, it is hard to conclude something. Given different protein concentrations used in the experiment, the authors should provide any kinds of value. For example, Sld3CBD-CDC45 shows weaker DNA binding than Sld3CBD alone (line 231). Is this true (or reproducible)? It is hard to conclude without any quantification.

We have repeated the EMSA assay four or more times with different rods of overexpression, purification and DNA synthesis, indicating that the EMSA assay is reproducible. In this revision, we changed the DNA stain and adjusted the ratio between the protein and ssDNA with increasing concentrations. The smeared bands of ssDNA with Sld7–Sld3ΔC–Cdc45 or Sld7–Sld3ΔC exhibit enhanced discernibility, and the ssDNA bands are intense enough for grayscale calculations (Figure 4 in the second revised version). We used a series of t-tests to confirm a significantly ssDNA residual level between Sld3CBD–Cdc45 to Sld3CBD, Sld7–Sld3ΔC–Cdc45, and Sld7–Sld3ΔCS (t-test, ****: P<0.0001). We also carefully controlled the sample amount in the EMAS assay and described it in the [Methods].

Moreover, in this EMSA assay (in Figure 4), the authors suggest that the disappearance of ssDNA bands corresponds with the binding of the protein to the DNA. However, it is also possible that the DNA is degraded. It is very important to show the band of protein-DNA complexes on the gel (a whole gel, not the parts of the gel shown in Figure). Why did the authors use this "insensitive" assay using SyberGreen, not radio-labelled ssDNA?

In this revision, we added a negative control of no ssDNA-binding by using ssARS1-3_3 for all protein samples (Sld3CBD, Sld3CBD–Cdc45, Sld7–Sld3ΔC–Cdc45 and Sld7–Sld3ΔC), which were the same rod of expression and purification for bound to ssARS1s (ssARS1-2 and ssARS1-5) (Figure 4), showing that the disappearance of ssDNA bands is caused by binding to proteins, not degradation. Moreover, this time, by changing the DNA stain and increasing the concentration of the samples, the smeared ssDNA bands exhibit enhanced discernibility in the high molecular weight regions when mixed with Sld7–Sld3ΔC–Cdc45 or Sld7–Sld3ΔC, whereas no bands appeared in the NC (ssARS1-3_1). The positions of smeared ssDNA bonds correspond to those of protein in the protein-stain pages, indicating that ssARS1 were complexed with proteins. Following the comment, we show all bands on the gel in Figure 4 and Supplementary Figure 14. Compared to Sld7–Sld3ΔC–Cdc45 or Sld7–Sld3ΔC, Sld3CBD and ssDNA bonds could not be observed because the pI value of Sld3CBD, which affects the entry of the samples into the gel.

We agree that using radio-labelled ssDNA can obtain a sensitive binding assay. However, current laboratory constraints did not allow us to use radio-labelled ssDNA. Furthermore, considering the characteristics of our target proteins, Sld3CBD, Sld3CBD–Cdc45, Sld7–Sld3ΔC–Cdc45, and Sld7–Sld3ΔC, we planned to perform the binding assay in a more natural state without any modifications, labelling or linkers. Additionally, we have attempted to use ITC experiments but failed in the measurements. Presumably, the conformational flexibility of Sld7-Sld3-Cdc45 and Sld7-Sld3 caused a thermodynamic anomaly.

Minor points:(1) Line 215, 80b: This should be "80 nucleotides(nt)". Throughout the text, nucleotides is better than base to show the length of ssDNAs.

Thank you for your comments. We modified these words throughout the text.